

# Trends in the incidence, prevalence and years lived with disability of facial fracture at global, regional and national levels from 1990 to 2017

Jin Wu[*], Anjie Min[*], Weiming Wang and Tong Su

Department of Oral and Maxillofacial Surgery, Center of Stomatology, Xiangya Hospital of Central South University, Changsha, Hunan, China
Research Center of Oral and Maxillofacial Tumor, Xiangya Hospital of Central South University, Changsha, Hunan, China
Institute of Oral Cancer and Precancerous Lesions, Central South University, Changsha, Hunan, China
[*] These authors contributed equally to this work.

## ABSTRACT

**Background**. Facial fracture is one of the most common injuries globally. Some types of facial fractures may cause irreversible damage and can be life-threatening. This study aimed to investigate the health burden of facial fractures at the global, regional, and national levels from 1990 to 2017.

**Methods**. Facial fracture data, including the incidence, prevalence, and years lived with disability (YLDs) from 1990 to 2017, were obtained from the Global Burden of Disease study. We calculated the estimated annual percentage changes (EAPCs) to assess the changes of facial fractures in 195 countries or territories and 21 regions.

**Results**. From 1990 to 2017, the change in cases of facial fracture incidence was 39% globally, while the age-standardized incidence rate showed a downtrend with an EAPC of 0.00. Syria experienced a ten-fold increase in incidence cases with an EAPC of 9.2, and this condition is largely responsible for the global health burden of facial fractures. The prevalence and YLDs showed a similar trend worldwide as the incidence. Additionally, we found that the incidence, prevalence, and YLDs showed a discrepancy among various age groups with a gradual change of proportion over the past 28 years. The age-standardized rates (ASRs) of facial fractures were nearly twice for male than those for female from 1990 to 2017.

**Conclusions**. EAPC showed a correlation with the ASRs of facial fractures and had no relationship with socio-demographic index. The proportion of children and elderly suffering from facial fractures slightly changed with time. The ratio of facial fractures between males and females was 2:1. These findings suggest that more targeted and specific strategies based on age and gender should be established in various countries and regions.

Corresponding authors
Weiming Wang,
wwmasly10@csu.edu.cn
Tong Su, sutong@csu.edu.cn,
sutongs@163.com

## INTRODUCTION

Injuries place huge burden on all populations worldwide, leading to high morbidity or mortality, regardless of age, gender, or geographical region (*Salomon et al., 2015*). Approximately 25% of all injuries reported in the National Trauma Data Bank involve the face (*Choi, Lorenz & Spain, 2020*). Facial fracture is a predominant cause of morbidity in the United States (*VandeGriend, Hashemi & Shkoukani, 2015*), because among the body parts, the face is the most exposed part, and it lacks protection, leading to the fragility of facial bones. Further, facial nerves and muscles responsible for central conduction, sensations, expressions, and eye movements are positioned near the facial bones (*Plaisier et al., 2000*). Therefore, fracture of face bones can result in death or inconvertible sequelae such as intracranial injury and severe psychosocial disorders (*Choi, Lorenz & Spain, 2020*; *Clavijo-Alvarez et al., 2012*; *Krishnan & Rajkumar, 2018*). Face fracture sites are multiple including frontal bone, nasal bone, zygomatic bone, maxilla and mandible, among which mandibular fracture was reported to account for the most common anatomic sites (*Lee, 2012*). Specifically, the distribution of anatomic locations in facial fractures showed a gradual alteration tendency from 2005 to 2010 according to a regional survey (*Roden et al., 2012*). Regarding to gender, major facial fractures are recorded in males, especially for those who aged between 16 and 30 years (*Haug, Prather & Indresano, 1990*). The most common causes of face trauma are assault, fall, and motor vehicle collision based on an epidemiology study of facial fractures in the United States (*Erdmann et al., 2008*). According to different era, geographic, socioeconomic, and cultural factors, the order of these three leading frequent causes may change (*Avansini Marsicano et al., 2019*; *Haug, Prather & Indresano, 1990*; *Simsek et al., 2007*). Based on numerous literatures about facial fractures, the distribution of the etiology and pattern of facial fractures were mostly investigated (*Alam et al., 2019*; *Fasola, Obiechina & Arotiba, 2003*; *Montovani et al., 2006*; *Thoren et al., 2009*). Though a recent study of facial fractures has analyzed the Global Burden of Disease(GBD) 2017 study (*Lalloo et al., 2020*), they focused on the causes of facial fractures globally and referred briefly to the incidence, prevalence, and years lived with disability (YLDs) of facial fractures. In the present study, we investigated the incidence, prevalence, and YLDs of facial fractures among 195 countries or territories and 21 regions by calculating the change in cases (CIC) and estimated annual percentage changes (EAPCs) from 1990 to 2017 to evaluate the changing trends of facial fractures and analyze the correlation between age-standardized rates (ASRs) or socio-demographic index (SDI) and EAPC. Furthermore, we discussed the changing trends in the age and gender distribution of facial fractures from 1990 to 2017.

The GBD study contains the data of 354 diseases and injuries in 195 countries or territories around the world including information about facial fractures, providing an opportunity for the exhaustive estimation of the distribution, burden, and trends of facial fractures in various countries and regions. GBD study produces the incidence, prevalence and years lived with disability to quantify the health loss for non-fatal disease such as facial fractures, thus contributing to the comparison of facial fractures with other health-damaging injuries or diseases. Moreover, GBD covers the distribution of facial fractures

among gender, age, SDI, region and country across a range of years, showing a changing trend of facial fractures. Indeed, many injuries or diseases showed a specific pattern with age or gender and may correlate with the social development. Therefore, GBD study which includes global data and cross-country comparisons might prompt current understanding of complex and multifactorial diseases such as facial fractures. In this study, we analyzed the GBD 2017 data of facial fractures and provided empirical evidence for policy makers in various countries or territories to adequately utilize the limited medical resources and formulate more targeted policies according to corresponding social development level, age distribution and gender difference in order to protect people from suffering facial fractures.

## MATERIALS & METHODS

### Study data sources

Data sources for the disease burden of facial fractures were collected using the GHDx (Global Health Data Exchange) online data source query tool (http://ghdx.healthdata.org/gbd-results-tool). Various indexes like "location", "year", "sex", "cause" and "age" can be selected in the query tool according to specific study objectives. The official GBD website provides a detailed instruction on the general methods applied for GBD 2017 (http://www.healthdata.org/gbd/). In this study, we obtained data on the annual number of cases for the incidence, prevalence, and YLDs of facial fractures at all-age levels and their corresponding ASRs including the 95% uncertainty interval (uncertainty interval given by GBD databases represents confidence interval) from 1990 to 2017. The three measurements including incidence, prevalence and YLDs collectively indicate the global burden of facial fractures. YLDs were assessed by multiplying the number of people living with facial fractures and disability weight that qualifies the magnitude of health loss owing to facial fractures (*Salomon et al., 2015*), so that providing an objective and comprehensive way to evaluate the loss of healthy life years in association with facial fractures. Standardization is essential for comparing the age distribution of various groups or the same group with different age composition over time. Age-standardized population in the GBD was calculated using the GBD world population age standard, which was exclusively explained in the supplementary appendix of published article by GBD 2017 collaborators. Also, we acquired the SDI of 194 countries or territories in 2017 from the official GBD website. SDI is a composite indicator that includes income per capita, average educational years and total fertility rate among individuals aged over 15 years. The calculation of SDI score in the GBD study was elaborated in the study by *Degenhardt et al. (2018)*, and we classified the SDI into the following 5 categories: low, low-middle, middle, high-middle and high. All data originated from GBDx are available for non-commercial users to share and modify via the Open Data Commons Attribution License.

### Statistical analysis

We analyzed the incidence, prevalence and YLDs of facial fractures worldwide by country, region, SDI, sex and age. The change in cases of incident or prevalent facial fractures and change in years lived with facial fractures in 195 countries or territories were calculated

to show the changing trends of facial fractures from 1990 to 2017. Furthermore, EAPC was calculated as previously described (*Advani, 2004*), using linear regression model based on the logarithm of the ASRs (*Kim et al., 2000*). Specifically, it is assumed that Y = $\alpha + \beta X + \varepsilon$, where Y represents ln (ASR), X means calendar year, and $\varepsilon$ refers to error term. Thus, annual percentage change and its 95% confidence interval are estimated through EAPC=100*(exp ($\beta$)-1). It has been shown that when the EAPC and lower limit of confidence interval are positive, ASR shows an increasing trend. Otherwise, if the EAPC and upper limit of the confidence interval are negative, the ASR shows a downward trend (*Deng et al., 2020*). World maps including 195 countries were drew to show the ASRs of facial fractures in 2017, change in number and EAPC in ASRs of facial fractures over the past 28 years visually. Moreover, we evaluated the relationship between EAPCs and ASRs in 1990, SDI in 2017 in different countries, aiming to investigate the potential social factors affecting EAPCs. The final output downloaded from GHDx is in comma-separated values (CSV) format. All statistics were imported into Excel files and performed using R program 3.5.3 (R Foundation for Statistical Computing, Vienna, Austria) or SPSS 25.0 (IBM Corporation, New York, USA). A *p* value of less than 0.05 was considered statistically significant. This study complies with the Guidelines for Accurate and Transparent Health Estimate Reporting recommendations.

## RESULTS

### Analysis of facial fracture incidence worldwide

From 1990 to 2017, the global incident cases of facial fractures rose from 5,405,814 to 7,538,663, increasing by 39.45%. Conversely, the EAPC of age-standardized incidence rate (ASIR) was 0.00 (−0.20 to 0.10), showing a downtrend, and the ASIR of facial fractures decreased from 100.47 per 100,000 persons to 98.47 per 100,000 persons over 28 years (Table 1). In addition, the incidence of facial fractures was 5,009,249 (4,113,772–6,093,590) in males, which was twice more than those in females, thus supporting the trend of ASIR.

In 2017, a high number of facial fractures incident cases was recorded in India (1,127,438.84), China (1,104,811.30), and USA (432,104.19), whereas a few incident cases were recorded in Northern Mariana Islands (36.68), Dominica (41.64), and American Samoa (43.22). Syria recorded the highest (588.34/100,000 people) ASIR in 2017, whereas Indonesia had the lowest ASIR (30.40/100,000 people) among the 195 countries or territories (Fig. 1A). Moreover, the ASIR of facial fractures was highest in Central Europe (310.03) among 21 regions but lowest in Southeast Asia (49.03, Table 1). Furthermore, the ASIR of facial fractures displayed a specific pattern with the various SDI values of 21 regions in the 2017 data. The regions in which the SDI was approximately 0.5 had low ASIR of facial fractures, while those with SDI near 0.8 presented a high ASIR (Fig. 2).

From 1990 to 2017, the incident cases of facial fractures increased in 159 countries or territories, whereas the EAPC was negative in 48 countries. As shown in Table 1, the greatest increase of percentage change in incident cases of facial fractures was observed in Oceania (157.89%), whereas the most prominent decrease of CIC was observed in Central Europe (−12.69%). High-income North America recorded the lowest EAPC of −1.87, whereas

**Table 1  The incidence of facial fractures, and its temporal trends from 1990 to 2017.**

| Characteristics | 1990 | | 2017 | | 1990-2017 | |
|---|---|---|---|---|---|---|
| | Incident cases No. ×10³ | ASIR per 100,000 No. | Incident cases No. ×10³ | ASIR per 100,000 No. | CIC No.(%) | EAPC No. |
| Global | 5405.81 | 100.47 | 7538.66 | 98.47 | 39.45 | 0.00 |
| **Sex** | | | | | | |
| Male | 3627.63 | 132.86 | 5009.25 | 130.14 | 38.09 | −0.03 |
| Female | 1778.19 | 67.15 | 2529.41 | 66.16 | 42.25 | −0.14 |
| **Socio-demographic index** | | | | | | |
| High SDI | 1630.33 | 175.66 | 1709.50 | 157.54 | 4.86 | −0.50 |
| High-middle SDI | 1428.97 | 127.38 | 1749.21 | 127.83 | 22.41 | 0.20 |
| Middle SDI | 909.18 | 57.87 | 1511.64 | 71.92 | 66.26 | 1.01 |
| Low-middle SDI | 749.74 | 72.96 | 1456.72 | 85.58 | 94.30 | 0.50 |
| Low SDI | 674.05 | 98.35 | 1089.16 | 86.27 | 61.59 | −0.60 |
| **Region** | | | | | | |
| Andean Latin America | 28.20 | 73.69 | 38.20 | 62.54 | 35.47 | −0.08 |
| Australasia | 51.20 | 264.17 | 77.13 | 290.95 | 50.62 | 0.42 |
| Caribbean | 18.52 | 52.95 | 33.50 | 72.16 | 80.84 | 1.64 |
| Central Asia | 122.39 | 173.15 | 154.20 | 167.19 | 25.99 | −0.39 |
| Central Europe | 387.03 | 315.87 | 337.91 | 310.03 | −12.69 | −0.06 |
| Central Latin America | 103.47 | 67.07 | 164.81 | 65.28 | 59.28 | 1.10 |
| Central Sub-Saharan Africa | 46.46 | 87.26 | 99.55 | 83.74 | 114.29 | −1.49 |
| East Asia | 684.58 | 52.12 | 1157.96 | 77.53 | 69.15 | 1.60 |
| Eastern Europe | 607.99 | 274.19 | 532.37 | 267.97 | −12.44 | 0.05 |
| Eastern Sub-Saharan Africa | 317.55 | 169.01 | 368.28 | 97.68 | 15.98 | −1.57 |
| High-income Asia Pacific | 247.26 | 150.69 | 259.95 | 157.77 | 5.13 | 0.11 |
| High-income North America | 515.19 | 189.25 | 481.47 | 131.42 | −6.55 | −1.87 |
| North Africa and Middle East | 368.77 | 105.86 | 783.03 | 126.96 | 112.33 | 1.00 |
| Oceania | 3.73 | 57.65 | 9.61 | 75.33 | 157.89 | 0.66 |
| South Asia | 762.17 | 71.09 | 1443.65 | 81.63 | 89.41 | 0.38 |
| Southeast Asia | 206.85 | 43.37 | 329.18 | 49.03 | 59.14 | 0.34 |
| Southern Latin America | 65.11 | 130.99 | 95.91 | 149.05 | 47.31 | 0.45 |
| Southern Sub-Saharan Africa | 52.07 | 99.70 | 67.96 | 87.02 | 30.53 | −0.47 |
| Tropical Latin America | 90.47 | 62.10 | 172.31 | 77.30 | 90.45 | 1.24 |
| Western Europe | 581.33 | 159.71 | 616.04 | 155.02 | 5.97 | −0.29 |
| Western Sub-Saharan Africa | 145.48 | 77.52 | 315.65 | 74.96 | 116.97 | −0.16 |

**Notes.**

ASIR, age-standardized incidence rate; SDI, socio-demographic index; YLDs, years lived with disability; EAPC, estimated annual percentage change.

Caribbean recorded the highest EAPC of 1.64. In addition, ASIR of facial fractures showed an average annual percentage change of less than zero in regions with high and low SDI quintiles (Fig. 3A). The trends of facial fractures over 28 years are presented in Fig. 4A. An obvious decline of ASIR was observed between 1995 and 2000 in high SDI quintile, and became stable subsequently. Additionally, a significant correlation was observed between

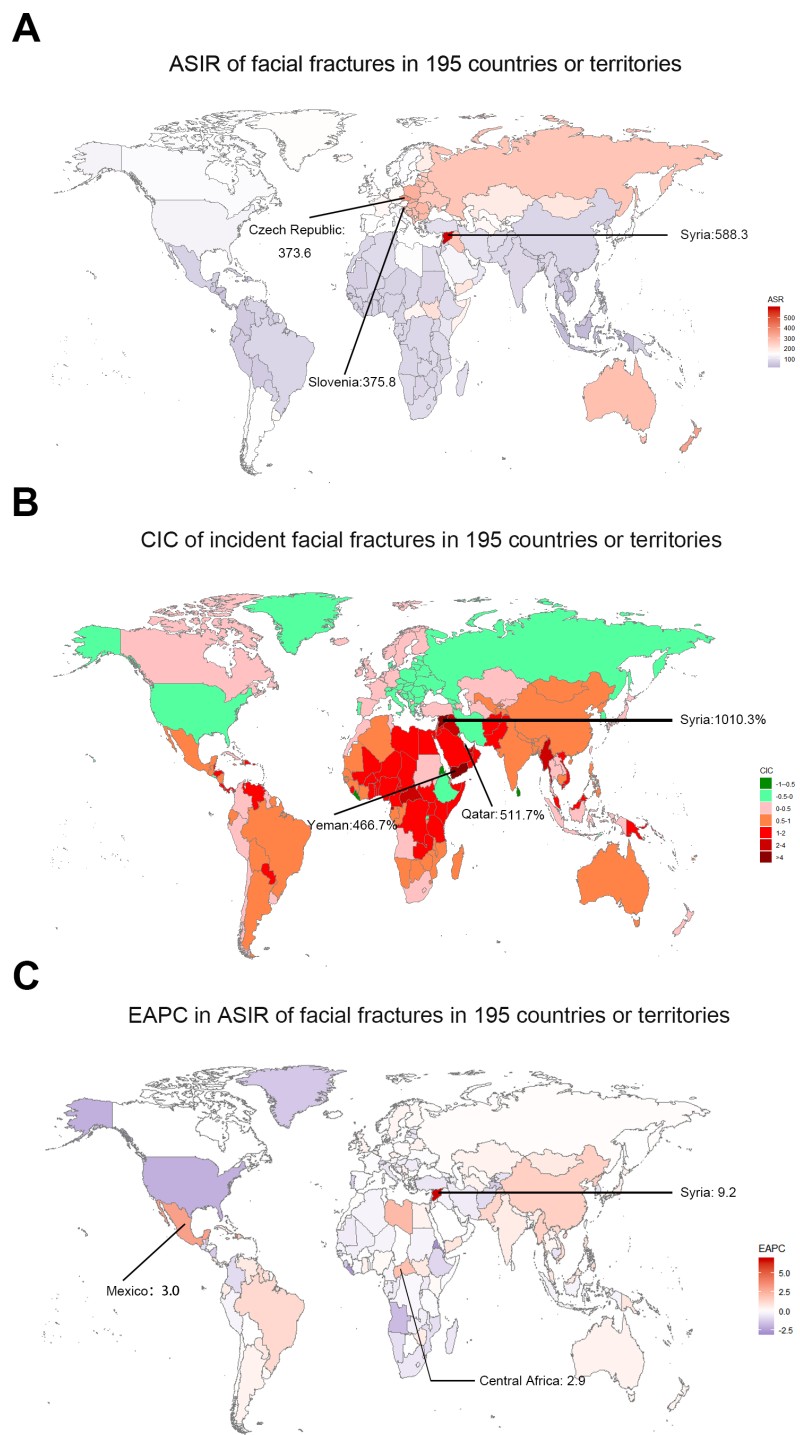

**Figure 1 The global incidence burden of facial fractures between 1990 and 2017 in 195 countries or territories.** (A) The ASIR of facial fractures in 2017. (B) The relative change in incident cases of facial fractures between 1990 and 2017. (C) The EAPC in ASIR of facial fractures from 1990 to 2017. The color shade presents the level of ASIR of facial fractures in 2017, percentage CIC of incident facial fractures, and EAPC in ASIR of facial fractures between 1990 and 2017 among 195 countries or territories. Warm color tone presents a high level while cold tone a lower level. The top three locations with the highest ASIR, CIC and EAPC in ASIR of facial fractures are marked in the maps respectively. ASIR, age-standardized incidence rate; CIC, change in cases; EAPC, estimated annual percentage change.

Peer J

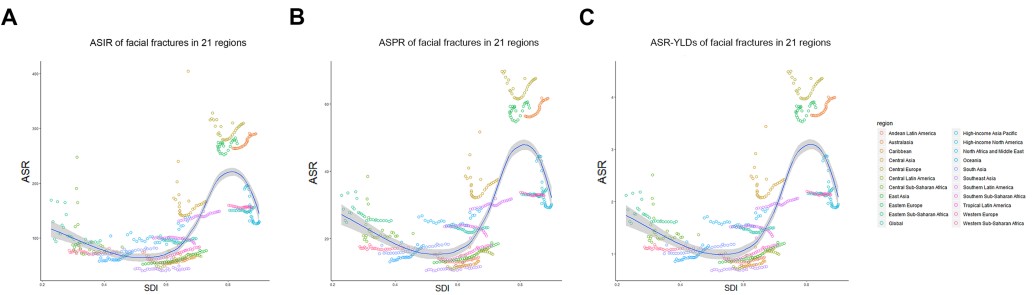

**Figure 2** **The association between age standardized rate of facial fractures and SDI among 21 regions between 1990 and 2017.** (A) ASIR and SDI. (B) ASPR and SDI. (C) Age-standardized YLDs rate and SDI. Each color represents a specific region. $X$-axis presents the SDI for each region and $Y$-axis presents corresponding ASIR (a), ASPR (b) and age-standardized YLDs rate (c) of facial fractures. Dots with the same color show the change trend of facial fractures in one region from 1990 to 2017. The fitted curve indicates that facial fractures varies widely with different SDI over 28 years. ASIR, age-standardized incidence rate; ASPR, age-standardized prevalence rate; YLDs, years lived with disability; SDI, socio-demographic index.

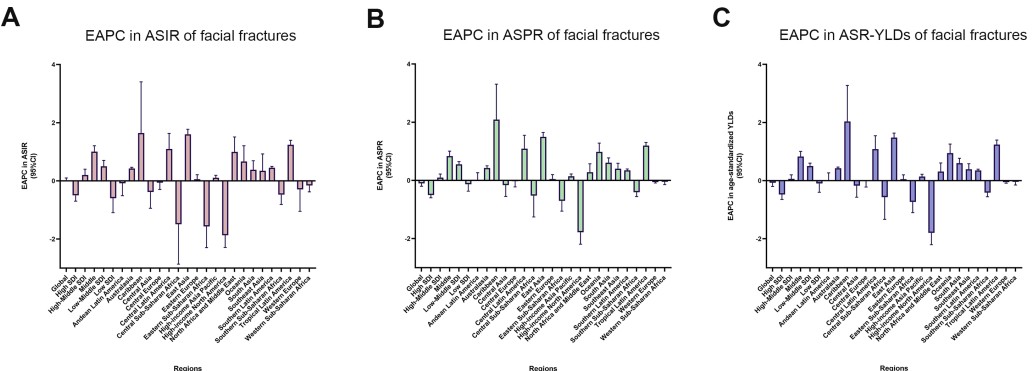

**Figure 3** **The EAPC of facial fractures in 21 regions from 1990 to 2017.** (A) The EAPC of ASIR. (B) The EAPC of ASPR. (C) The EAPC of age-standardized YLDs rate. $X$-axis presents 21 different regions and $Y$-axis presents the EAPC of ASIR (A), ASPR (B) and age-standardized YLDs rate (C) of facial fractures respectively. The positive value of EAPC indicates an increasing trend in corresponding ASR of facial fractures whilst the negative value shows a downtrend. The EAPC across 21 regions are shown in 95% confidence interval (CI). EAPC, estimated annual percentage change; ASIR, age-standardized incidence rate; ASPR, age-standardized prevalence rate; YLDs, years lived with disability; ASR, age-standardized rate.

EAPC and ASIR ($\rho = -0.3842$, $P < 0.0001$, Fig. 5A), while no association was found between EAPC and SDI ($\rho = 0.0036$, $P = 0.9603$, Fig. 5B).

## Analysis of facial fracture prevalence worldwide

During the last 28 years, global prevalence of facial fractures increased by 54.39% from 1,819,732 in 1990 to 1,178,636 in 2017. By contrast, the age-standardized prevalence rate (ASPR) decreased worldwide, with an EAPC of $-0.10$ ($-0.20$ to $0.10$). Similar to incidence, the fracture of face bones predisposed to occur in males (1,155,326 prevalent numbers), which was nearly twice more than that in females (664,406 prevalent numbers), and it was consistent with the prevalence rate after age standardization.

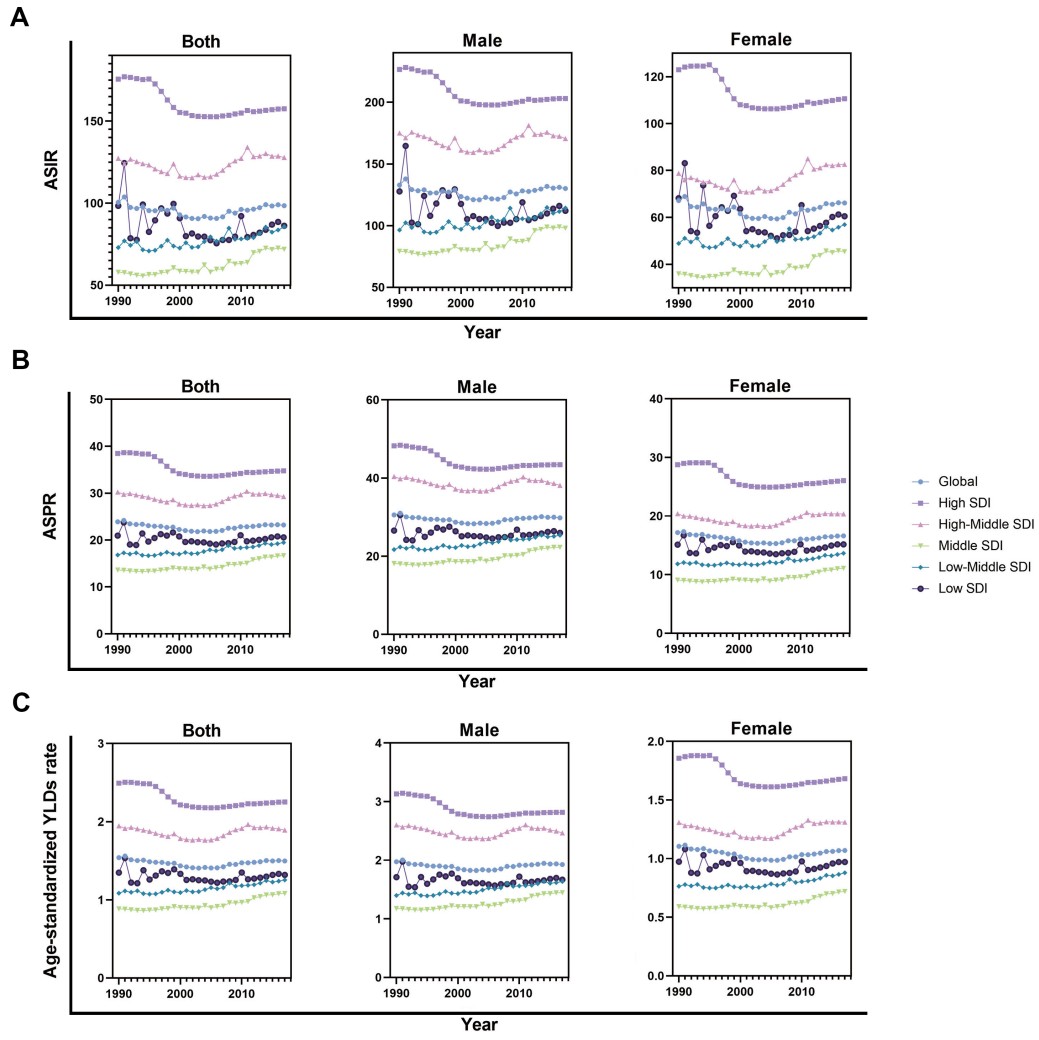

**Figure 4** **The change trends of age-standardized rate of facial fractures among distinct SDI quintiles and gender from 1990 to 2017.** (A) ASIR. (B) ASPR. (C) Age-standardized YLDs rate. The SDI of 21 regions are classified in quintiles. Each shape and color of dots present specific SDI quintile. The results are shown in both sexes, males and females. The curves show the different changing trend of facial fractures among distinct SDI quintiles in males, females and both sexes over last 28 years. ASIR, age-standardized incidence rate; ASPR, age-standardized prevalence rate; YLDs, years lived with disability; SDI, socio-demographic index; ASR, age-standardized rate.

Among the countries with various prevalence cases of facial fractures, the lowest ASPRs were recorded in Indonesia (8.7/100,000 people) and Mauritius (11.8/100,000 people, Fig. 6A). From 1990 to 2017, the ASPR of facial fractures decreased in 82 countries, in which Eritrea recorded a maximum decrease of 77.08% (Fig. 6B). As shown in Table 2, the number of prevalent facial fractures increased in most regions and decreased only in Eastern Europe, while ASPR showed a downward trend in 10 regions, in which high-income North America recorded the most prominent decrease of ASPR (−29.16%) with the lowest EAPC of −1.78. Otherwise, facial fractures prevalence showed an upward trend in 11 regions, in

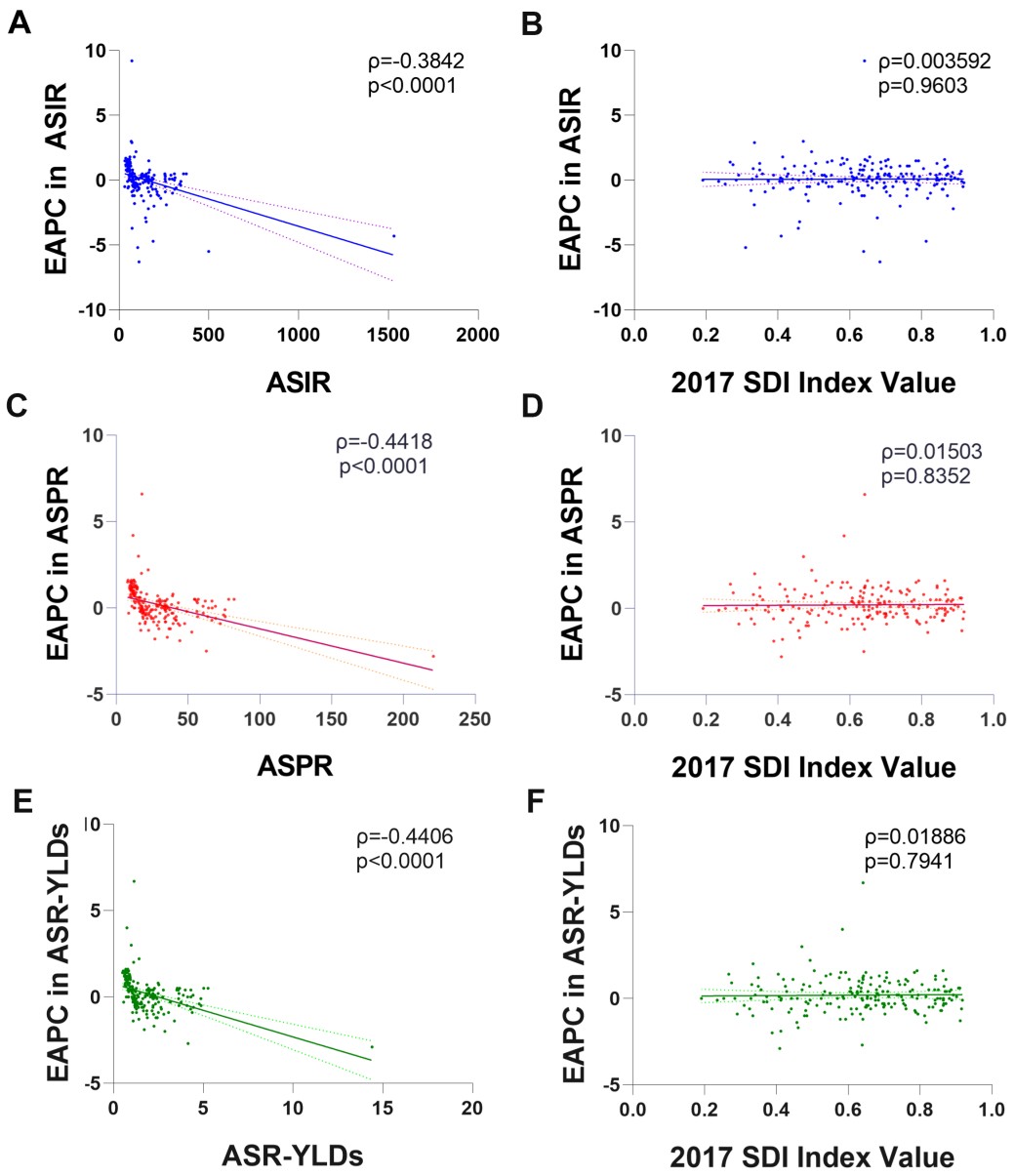

**Figure 5** **The correlation between EAPC and age standardized rate of facial fractures in 1990 as well as SDI in 2017 among 194 countries or territories.** (A) EAPC and ASIR. (B) EAPC and SDI in incidence. (C) EAPC and ASPR. (D) EAPC and SDI in prevalence. (E) EAPC and age-standardized YLDs rate. (F) EAPC and SDI in YLDs. The dots represent countries that were available on SDI data. The indices $\rho$ and $p$ values presented are obtained from Pearson correlation analysis. The EAPC shows a negative correlation with corresponding ASRs and not related with SDI among 194 countries or territories. EAPC, estimated annual percentage change; ASIR, age-standardized incidence rate; ASPR, age-standardized prevalence rate; YLDs, years lived with disability; ASR, age-standardized rate.

which Caribbean increased fastest (2.09) followed by East Asia (1.50). In comparison with the 1990 data, the ASPR of facial fractures decreased in high and low SDI quintiles in 2017. The age-standardized rates of prevalent facial fractures among five multiple SDI grades in

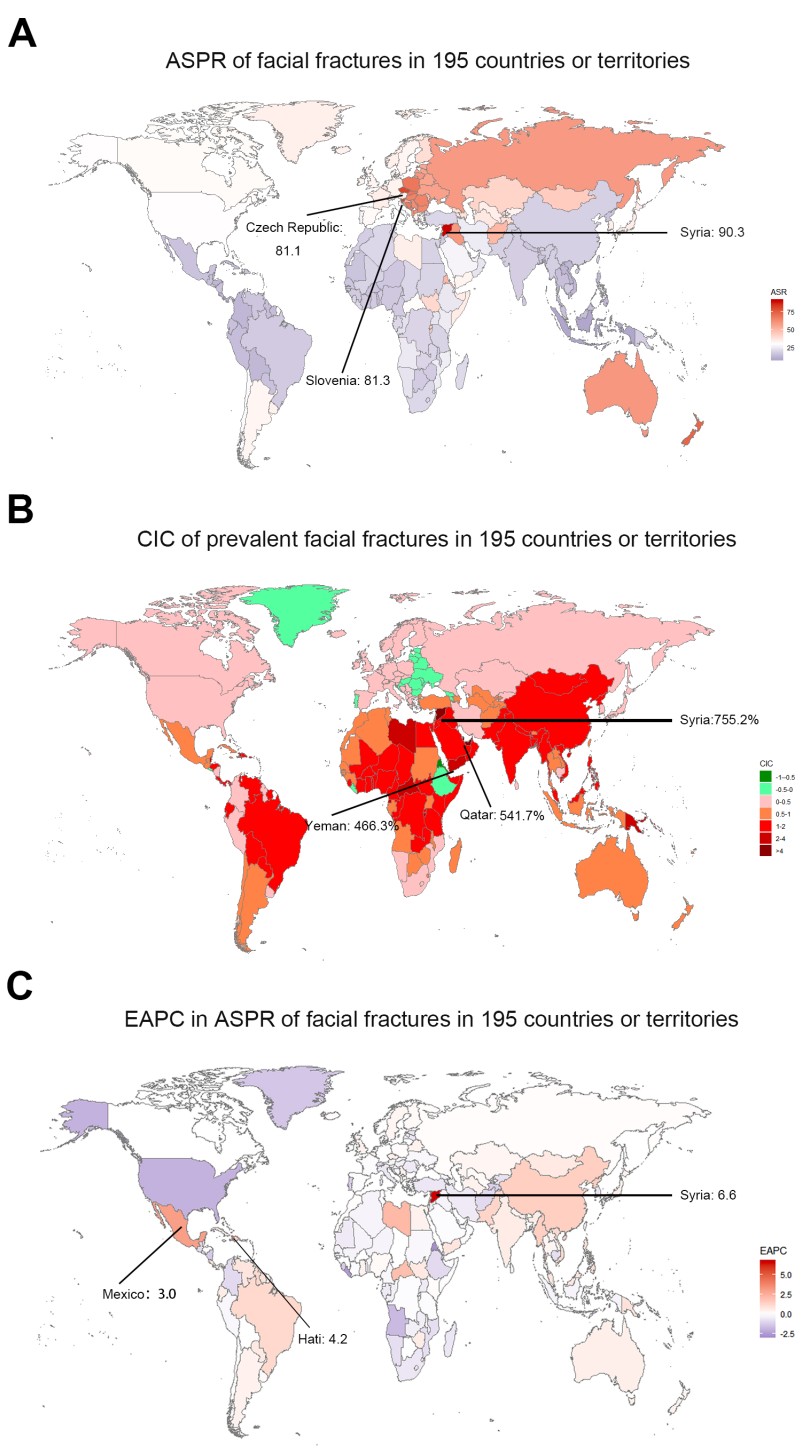

**Figure 6** **The global prevalence burden of facial fractures in 195 countries and territories.** (A) The ASPR of facial fractures in 2017. (B) The change in prevalent cases of facial fractures between 1990 and 2017. (C) The EAPC of facial fractures ASPR from 1990 to 2017. The color shade presents the level of ASPR of facial fractures in 2017, percentage CIC of prevalent facial fractures, and EAPC in ASPR of facial fractures between 1990 and 2017 among 195 countries or territories. Warm color tone presents a high level while cold tone presents a lower level. The top three locations with the highest ASPR, CIC of prevalent facial fractures and EAPC in ASPR are marked in the maps respectively. ASPR, age-standardized prevalence rate; CIC, change in cases; EAPC, estimated annual percentage change.

**Table 2   The prevalence of facial fractures, and its temporal trends from 1990 to 2017.**

| Characteristics | 1990 | | 2017 | | 1990–2017 | |
|---|---|---|---|---|---|---|
| | Prevalent cases No. ×10³ | ASPR per 100,000 No. | Prevalent cases No. ×10³ | ASPR per 100,000 No. | CIC No.(%) | EAPC No. |
| Global | 1178.64 | 23.87 | 1819.73 | 23.20 | 54.39 | −0.10 |
| **Sex** | | | | | | |
| Male | 754.57 | 30.56 | 1155.33 | 29.83 | 53.11 | −0.03 |
| Female | 424.07 | 17.18 | 664.41 | 16.62 | 56.67 | −0.15 |
| **Socio-demographic index** | | | | | | |
| High SDI | 397.01 | 38.45 | 475.20 | 34.75 | 19.69 | −0.50 |
| High-middle SDI | 323.55 | 30.21 | 453.56 | 29.27 | 40.18 | 0.10 |
| Middle SDI | 187.35 | 13.61 | 364.25 | 16.66 | 94.42 | 0.84 |
| Low-middle SDI | 147.76 | 16.82 | 300.83 | 19.45 | 103.59 | 0.55 |
| Low SDI | 119.94 | 20.93 | 219.79 | 20.56 | 83.24 | −0.13 |
| **Region** | | | | | | |
| Andean Latin America | 5.10 | 15.29 | 8.46 | 14.31 | 66.08 | 0.02 |
| Australasia | 11.73 | 56.38 | 19.52 | 61.75 | 66.47 | 0.43 |
| Caribbean | 3.76 | 11.64 | 8.43 | 17.53 | 124.18 | 2.09 |
| Central Asia | 24.35 | 38.58 | 33.55 | 37.62 | 37.78 | −0.16 |
| Central Europe | 91.50 | 69.67 | 92.39 | 67.62 | 0.97 | −0.02 |
| Central Latin America | 20.90 | 15.72 | 37.30 | 14.99 | 78.48 | 1.09 |
| Central Sub-Saharan Africa | 8.89 | 20.88 | 20.30 | 21.52 | 128.18 | −0.52 |
| East Asia | 153.80 | 12.87 | 324.20 | 18.72 | 110.79 | 1.50 |
| Eastern Europe | 143.33 | 59.29 | 139.75 | 57.94 | −2.50 | 0.05 |
| Eastern Sub-Saharan Africa | 51.18 | 32.89 | 75.00 | 25.62 | 46.54 | −0.70 |
| High-income Asia Pacific | 59.84 | 33.18 | 80.15 | 35.04 | 33.92 | 0.14 |
| High-income North America | 127.98 | 43.03 | 132.60 | 30.48 | 3.61 | −1.78 |
| North Africa and Middle East | 77.94 | 26.47 | 159.84 | 27.80 | 105.07 | 0.28 |
| Oceania | 0.69 | 13.17 | 1.96 | 18.24 | 182.88 | 0.98 |
| South Asia | 143.12 | 15.57 | 303.06 | 18.38 | 111.76 | 0.61 |
| Southeast Asia | 43.10 | 10.56 | 81.65 | 12.31 | 89.45 | 0.40 |
| Southern Latin America | 14.02 | 28.77 | 22.10 | 31.85 | 57.59 | 0.35 |
| Southern Sub-Saharan Africa | 10.38 | 23.55 | 14.51 | 20.21 | 39.83 | −0.41 |
| Tropical Latin America | 18.26 | 14.04 | 40.08 | 17.40 | 119.45 | 1.19 |
| Western Europe | 142.03 | 33.77 | 166.36 | 32.65 | 17.13 | −0.06 |
| Western Sub-Saharan Africa | 26.73 | 17.55 | 58.52 | 17.31 | 118.95 | −0.04 |

Notes.
   ASPR, age-standardized prevalence rate; SDI, socio-demographic index; CIC, change in cases; EAPC, estimated annual percentage change.

the past 28 years are shown in Fig. 3B. EAPC showed a negative correlation with the ASPR of facial fractures ($\rho = -0.4418$, $P < 0.0001$, Fig. 5C) but had no relationship with SDI ($\rho = 0.0150$, $P = 0.8352$, Fig. 5D).

 

**Table 3  The YLDs of facial fractures, and its temporal trends from 1990 to 2017.**

| Characteristics | 1990 | | 2017 | | 1990–2017 | |
|---|---|---|---|---|---|---|
| | YLDs No. ×10³ | Age-standardized YLDs rate (per 100,000) No. | YLDs No. ×10³ | Age-standardized YLDs rate (per 100,000) No. | CIC No.(%) | EAPC No. |
| Global | 76.51 | 1.54 | 117.4 | 1.50 | 53.44 | −0.07 |
| **Sex** | | | | | | |
| Male | 49.10 | 1.97 | 74.73 | 1.93 | 52.2 | −0.02 |
| Female | 27.41 | 1.10 | 42.67 | 1.07 | 55.67 | −0.05 |
| **Socio-demographic index** | | | | | | |
| High SDI | 25.59 | 2.49 | 30.33 | 2.25 | 18.52 | −0.48 |
| High-middle SDI | 20.93 | 1.95 | 29.16 | 1.89 | 39.32 | 0.06 |
| Middle SDI | 12.30 | 0.88 | 23.65 | 1.08 | 92.28 | 0.83 |
| Low-middle SDI | 9.65 | 1.08 | 19.57 | 1.25 | 102.8 | 0.50 |
| Low SDI | 7.84 | 1.35 | 14.29 | 1.32 | 82.27 | −0.10 |
| **Region** | | | | | | |
| Andean Latin America | 0.33 | 1.00 | 0.55 | 0.93 | 66.67 | 0.02 |
| Australasia | 0.76 | 3.66 | 1.25 | 4.00 | 64.47 | 0.42 |
| Caribbean | 0.25 | 0.76 | 0.54 | 1.13 | 116 | 2.04 |
| Central Asia | 1.58 | 2.49 | 2.18 | 2.43 | 37.97 | −0.17 |
| Central Europe | 5.86 | 4.48 | 5.86 | 4.36 | 0 | −0.02 |
| Central Latin America | 1.37 | 1.02 | 2.42 | 0.97 | 76.64 | 1.09 |
| Central Sub-Saharan Africa | 0.58 | 1.33 | 1.32 | 1.37 | 127.59 | −0.57 |
| East Asia | 10.09 | 0.84 | 20.98 | 1.22 | 107.93 | 1.48 |
| Eastern Europe | 9.21 | 3.83 | 8.91 | 3.74 | −3.26 | 0.05 |
| Eastern Sub-Saharan Africa | 3.36 | 2.12 | 4.87 | 1.63 | 44.94 | −0.73 |
| High-income Asia Pacific | 3.88 | 2.16 | 5.11 | 2.28 | 31.7 | 0.14 |
| High-income North America | 8.24 | 2.78 | 8.45 | 1.97 | 2.55 | −1.80 |
| North Africa and Middle East | 5.05 | 1.69 | 10.34 | 1.79 | 104.75 | 0.31 |
| Oceania | 0.05 | 0.85 | 0.13 | 1.17 | 160 | 0.95 |
| South Asia | 9.38 | 1.01 | 19.71 | 1.19 | 110.13 | 0.60 |
| Southeast Asia | 2.83 | 0.69 | 5.31 | 0.80 | 87.63 | 0.39 |
| Southern Latin America | 0.91 | 1.87 | 1.43 | 2.07 | 57.14 | 0.35 |
| Southern Sub-Saharan Africa | 0.68 | 1.52 | 0.94 | 1.30 | 38.24 | −0.41 |
| Tropical Latin America | 1.20 | 0.91 | 2.60 | 1.13 | 116.67 | 1.24 |
| Western Europe | 9.15 | 2.19 | 10.64 | 2.12 | 16.28 | −0.06 |
| Western Sub-Saharan Africa | 1.75 | 1.13 | 3.84 | 1.12 | 53.44 | −0.05 |

**Notes.**
YLDs, years lived with disability; SDI, socio-demographic index; CIC, change in cases; EAPC, estimated annual percentage change.

## Analysis of facial fracture YLDs worldwide

As shown in Table 3, the years lived with facial fractures were 117,402.03 years in 2017, which was 1.5-folds higher than those in 1990. Similarly, the age-standardized YLDs rate had an average EAPC of −0.07, indicating a downward trend. YLDs and age-standardized YLDs rates were both high in males and were twice higher than those in females.

The top three countries with high YLDs were the same as incidence and prevalence in 2017. Years living with disability owing to facial fractures were low in Marshall Islands (0.58), Northern Mariana Islands (0.60), and American Samoa (0.62, Fig. 7A). Similar to ASIR and ASPR, the highest age-standardized YLDs was recorded in Syria (5.90), whereas Indonesia had the lowest (0.56). The values of EAPC were negative in 76 countries from 1990 to 2017 (Fig. 7C). Eritrea recorded the lowest EAPC of −2.9 (−4.2 to 1.6), while Syria recorded the highest EAPC of 6.7 (4.1–9.3). At the regional level, the YLDs of facial fractures were high in East Asia (20.98), South Asia (19.71), and Western Europe (10.64), while age-standardized YLDs rate was higher in Central Europe (4.36), Australasia (4.00), and Eastern Europe (3.74, Table 3). Only in high and high-middle SDI quintile, the years lived with facial fractures increased by less than 50%. Analogously, the greatest increase of facial fractures YLDs was observed in middle SDI quintile (102.8%). The variation tendency of age-standardized YLDs over the past 28 years are presented in Fig. 4C. EAPC was negatively correlated with age-standardized YLDs ($\rho = -0.4406$, $P < 0.0001$, Fig. 5E) but not correlated with SDI ($\rho = -0.0189$, $P = 0.7941$, Fig. 5F).

### Age and gender distribution of incidence, prevalence, and YLDs of facial fractures

From 1990 to 2017, in all regions, the incident, prevalent cases, and YLDs in connection with facial fractures were mainly assembled between the age of 15 and 49 years, followed by the age group of 50–69 years. In addition, in both sexes, these indices showed upward trends over time in people aged over 50 years and downward trends in people under 14 years. Notably, a dynamic equilibrium was observed in people aged 15–49 years across 28 years (Fig. 8).

Both in 1990 and 2017, the incidence rate of facial fractures showed double peaks at the age of 20–35 years and over 80 years in males (Figs. 9A and 9B). The incidence rate in females showed a flat pattern in different age groups except for those over 80 years and presented a sharp increase of incidence rate (Figs. 9A and 9B). The prevalence and YLDs rate of facial fractures increased gradually with age and reached the peak at age over 80 years in both sexes (Figs. 9C–9F).

In both sexes, the age-standardized incidence rate, prevalence rate and years lived with facial fractures showed a relatively stable trend globally from 1990 to 2017 (Fig. 10A). Interestingly, the above parameters in males were always twice higher than those in females. In Syria, where the heaviest global burden of facial fractures was observed, these parameters showed an upsurge in 2011, and similarly the gender proportion between females and males remained approximately 1:2, though a sharp increase was observed since 2011 (Fig. 10B).

## DISCUSSION

Our analysis based on the GBD study displayed the latest worldwide patterns and the trends in the incidence, prevalence, and YLDs of facial fractures. From 1990 to 2006, the ASIR, ASPR, and age-standardized YLDs rate of facial fractures showed a slow downward trend globally. However, since 2007, among both females and males, the facial fracture burden

**A**

ASR-YLDs of facial fractures in 195 countries or territories

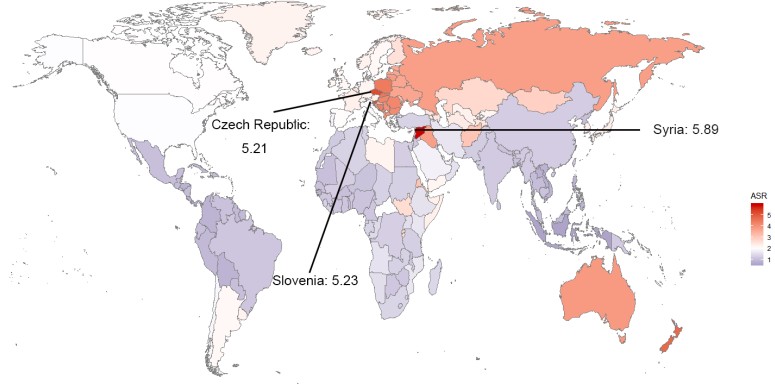

**B**

Change in years lived with facial fractures in 195 countries or territories

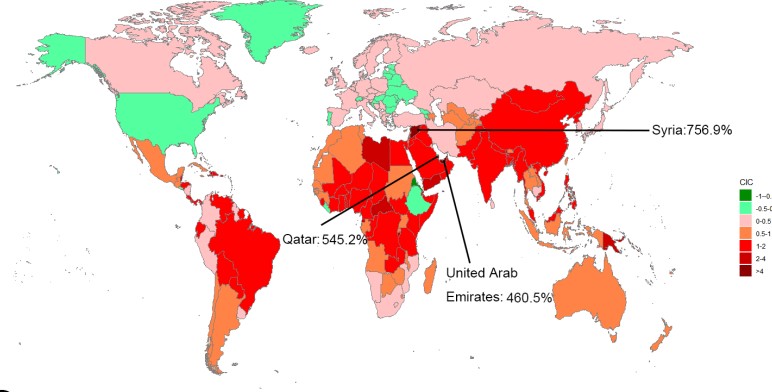

**C**

EAPC in ASR-YLDs of facial fractures in 195 countries or territories

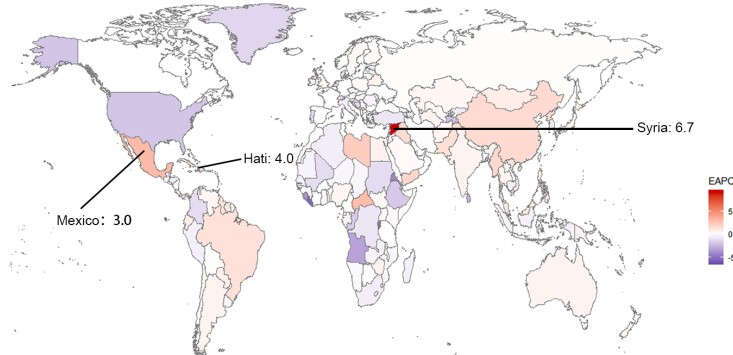

**Figure 7 The global YLDs burden of facial fractures in 195 countries and territories.** (A) The age-standardized YLDs rate of facial fractures in 2017. (B) The change years lived with disability owning to facial fractures between 1990 and 2017. (C) The EAPC of facial fractures age-standardized YLDs rate from 1990 to 2017. The color shade represents the level of age-standardized YLDs rate of facial fractures in 2017, percentage change in YLDs of facial fractures, and the EAPC in age-standardized YLDs rate of facial fractures between 1990 and 2017 among 195 countries or territories. Warm color tone presents high level while cold tone presents a lower level. The top three locations with the highest ASR-YLDs, percentage change in years lived with facial fractures and EAPC in ASR-YLDs are marked in the maps respectively. YLDs, years lived with disability; EAPC, estimated annual percentage change; ASR, age-standardized rate.

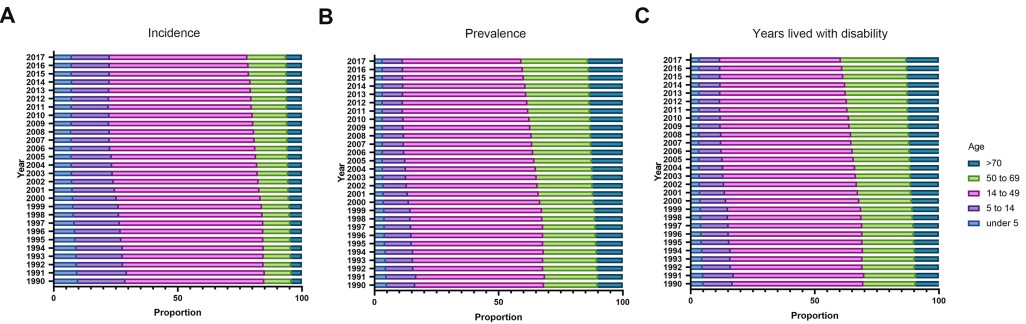

**Figure 8** **The proportion of different age groups in facial fractures by years between 1990 and 2017.**
(A) Incidence. (B) Prevalence. (C) Years lived with disability. Each color presents specific age group. The age distribution of facial fracture alters with the years advancing from 1990 to 2017.

increased slightly. The facial fracture burden was more skewed towards males than females worldwide. These findings will provide basis for policy makers to allocate medical resources reasonably and determine the underlying causes of facial fractures, thus decreasing the incidence rate of facial fractures. In addition, considering the various developing trends of facial fractures in different SDI quintiles, more targeted policy should be applied.

As shown in Fig. 2, the ASIR, ASPR, and age-standardized YLDs presented a similar and specific pattern according to the different SDIs. Regions with higher SDI showed higher ASRs than those with lower SDI. Motor vehicle collision, fall, and assault are the top three causes of facial fractures (*Avansini Marsicano et al., 2019*). Therefore, the regions with high SDI had high motor vehicle numbers and traffic flux, which increased the incidence risk of road accidents and facial fractures indirectly. However, the high and low SDI quintiles showed a negative EAPC, indicating that these regions showed a downward trend annually between 1990 and 2017. The use rates of seat belt in facial fractures increased by 3% (1990–1995), 8% (1996–2000), and 15% (2000–2004) (*VandeGriend, Hashemi & Shkoukani, 2015*). Cormier et al. also demonstrated the importance of using seat belts in the mitigation of facial injuries (*Cormier & Duma, 2009*). Hence, the increase in the use of seat belts, helmets, and advanced car models and the strict control of road traffic may contribute to the negative EAPC of facial fractures in regions with high SDI quintile (*Czerwinski et al., 2008*; *Erdmann et al., 2008*). The development and application of 3D printing technology has also been increasingly mature in the last three decades (*Liu et al., 2018*), thereby improving the cure rate of facial fractures in developed countries or territories. Regions with low SDI showed low age-standardized incidence rate, prevalence rate and years lived with disability, which may be attributed to the hysteretic medical level and insufficient diagnosis ability (*Erdmann et al., 2008*; *VandeGriend, Hashemi & Shkoukani, 2015*). Moreover, the probability of occurrence of motor vehicle collisions, as one of the main causes of facial fractures, is low in regions with low SDI (*Alam et al., 2019*). However, regions with middle and high-middle SDI showed high EAPCs, indicating a potential developing trend of facial fractures in the future. Considering the economic growth in these regions, the numbers of motor vehicles increased, whereas

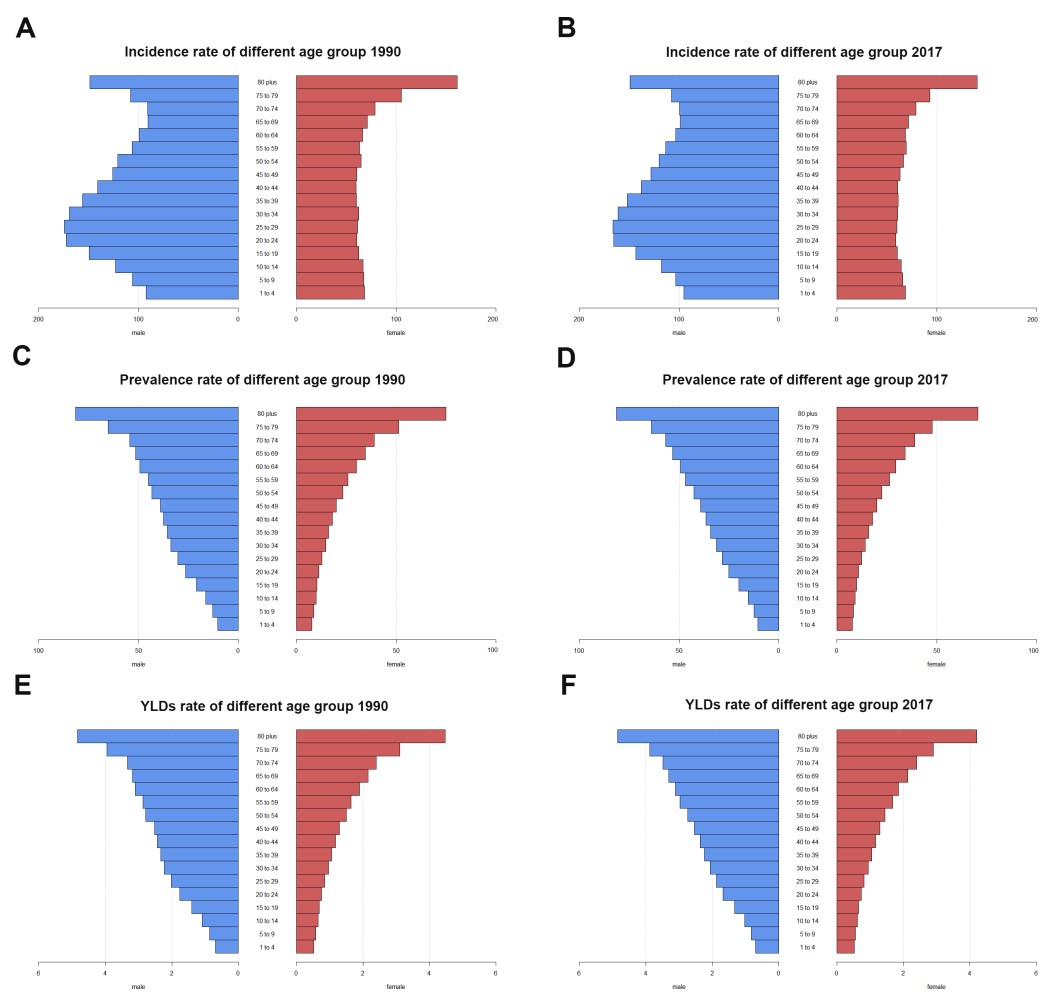

**Figure 9 The rate of facial fractures between males and females among different age groups in 1990 and 2017.** (A) Incidence rate. (B) Prevalence rate. (C) Years lived with disability rate. Blue bar indicates males and red bar indicates females. *X*-axis presents rates of facial fractures and *Y*-axis presents different age brackets. Left column presents rates of facial fractures in 1990 while the right presents in 2017. Incidence rate shows doublets in males and singlet in females in 1990 and 2017. Prevalence rate and years lived with disability rate of facial fractures increase with aging. YLDs, years lived with disability.

the corresponding road laws and regulations did not keep the path, thus increasing the incidence, prevalence, and YLDs of facial fractures. The economic growth may also prompt the advancement of medical equipment to raise the diagnosis numbers of facial fractures.

The EAPC of ASIR, ASPR, and age-standardized YLDs had no correlation with SDI, because the fracture mechanisms and risk factors of facial fractures are irrelevant with the SDI or regions. Conversely, EAPC showed negative correlations with the ASRs of incidence, prevalence, and YLDs. Facial fractures aggravated the health and financial burden globally. The total hospitalization charges for facial fracture in the United States were $1.06 billion in 2008 (*Nalliah et al., 2013*). Accordingly, the government may implement all kinds of effective measures such as punishing the drunk drivers strictly and propagating the use

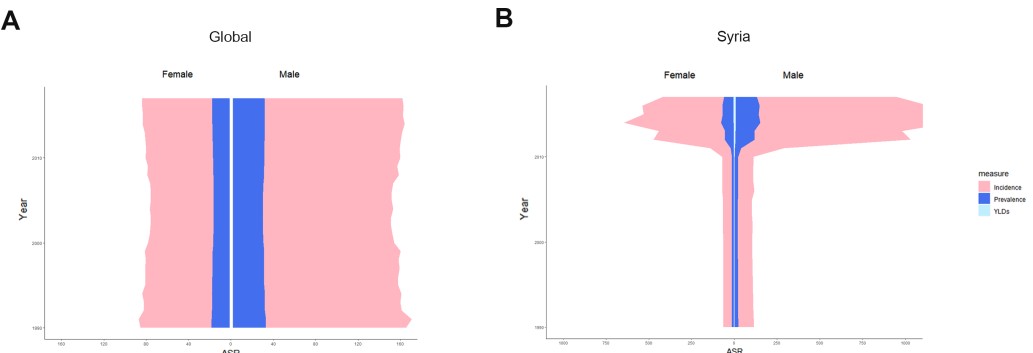

**Figure 10  The trends of age standardized rate of facial fractures in global and Syria by gender from 1990 to 2017.** (A) Global. (B) Syria. Each color presents specific ASR that pink indicates ASIR, dark blue the ASPR and pale blue the age-standardized YLDs rate. *X*-axis presents ASRs in females (left) and males (right), and *Y*-axis the different years. The ratio of males: females tends to be approximately 2:1 worldwide and facial fractures in Syria showed an upsurge in 2011. ASR, age-standardized rate; ASIR, age-standardized incidence rate; ASPR, age-standardized prevalence rate; YLDs, years lived with disability.

of seat belts and helmets to decrease the health and financial burden of facial fractures. Therefore, EAPC is shown to be inversely related to ASIR, ASPR, and age-standardized YLDs in 194 countries or territories.

As shown in Fig. 8, facial fractures were less frequent in children, and the ASPR of patients with facial fractures increased equally with age (Fig. 9), supporting the results of *Imahara et al. (2008)*. Family care, reduced outdoor activities, and increased observations (*Montovani et al., 2006*) on children resulted in the low incidence, prevalence, and YLDs. In addition, the high bone elasticity and thick soft-tissue layer strengthen the resistance of children against facial fractures, especially maxillary and Le Fort fractures, because the frontal, ethmoid, sphenoid, and maxillary sinuses of children tend to be small and lack pneumatization (*Chan et al., 2004*; *Thoren et al., 2009*). Over 28 years, the proportion of facial fractures in various age groups was gradually altered, the groups aged under 15 years showed a decline of ASRs from 1990 to 2017, whereas those aged over 50 years showed an upward trend. This change can be linked to the demographic alteration worldwide. The population of the world is aging gradually because of the increase in life expectancy (*Liu et al., 2019*), which may lead to the ascending tendency of facial fractures incidence, prevalence, and YLDs among senior citizens, while the low fertility rates of many regions (*Nogami et al., 2019*) and the growing attention to children safety may lead to the downward trends of pediatric facial fractures globally. As shown in Fig. 9, both in males and females, the prevalence rate and YLDs rate increased with aging in 1990 and 2017. Falls are the most common cause of facial fracture among elderly patients (*Atisha et al., 2016*; *Erdmann et al., 2008*). The high risk of facial fractures among the elderly patients might have several causes, including the deficiency of the balance and strength while moving or resisting crashing and the tendency to suffer from various age-associated comorbidities, such as osteoarthritis and visual impairment (*Atisha et al., 2016*). The most commonly injured sites in elderly patients are the mandibular and nasal bones (*Wade, Hoffman & Brennan, 2004*),

and associated injuries from facial fractures occur often and are severe in geriatric patients, leading to a high death rate (*Toivari et al., 2016*). The elderly individuals will account for 20% of the population in the United States of America in 2030, and a similar figure can be recorded in other developed countries (*Vlavonou, Nguyen & Toure, 2018*). Therefore, effective measures should be implemented to immediately prevent elderly people from suffering from facial fractures.

Interestingly, unlike females, the incidence rate of facial fractures in males also peaked at people aged between 25–29 years both in 1990 and 2017. Moreover, as shown in Fig. 10, the ASIR, ASPR, and age-standardized YLDs were always twice higher in males than those in females from 1990 to 2017. This finding may be related to the "high risk" recreational activities among young males, such as bicycling and sports (*Plawecki et al., 2017*) and that men are involved in driving (*Montovani et al., 2006*). In an Indian survey, the increased incidence of facial injuries and fractures among young men was explained by the reluctance to use helmets, exceeding speed limits, lack of tolerance, and increasing competition (*Subhashraj, Nandakumar & Ravindran, 2007*). Syria recorded the highest ASRs of facial fractures worldwide. Therefore, we analyzed the changing trend of ASIR, ASPR, and age-standardized YLDs in males and females since 1990 in Syria, and the data showed a sharp increase of ASIR, ASPR, and age-standardized YLDs in 2011. We considered an association with the explosion during the Syrian war in 2011 (*Hayani, Dandashli & Weisshaar, 2015*). Moreover, the ratio between males and females did not substantially change before and after 2011.

## CONCLUSIONS

In conclusion, the incident cases, prevalent cases, and YLDs of facial fractures increased worldwide. By contrast, the ASRs showed downtrends globally. EAPC showed a correlation with the ASRs of facial fractures and was hardly associated with SDI. The ratio between males and females approached 2:1. Besides, the proportion of children and elderly suffering from facial fractures slightly changed with time. Facial fractures occurred more in young men aged between 25–29 years and in the elderly aged over 80 years in both sexes.

To our best knowledge, this study was the first to systematically investigate the changing trends of incidence, prevalence, and YLDs in facial fractures from 1990 to 2017 at the global, national, and regional levels. However, this study has some limitations. The accuracy of results depended on the quality and quantity of GBD data. For instance, the method utilized in the GBD study cannot provide access to cover all districts worldwide, and the opportunity to be diagnosed with facial fractures was not equal between developed and less developed countries because of differences in specialized medical care and imaging resources. Besides, in this text, we focused solely on the changing tendency of facial fractures in incidence, prevalence, and YLDs and did not analyze the trends of causes leading to facial fractures at the global, national, and regional levels from 1990 to 2017, which could be discussed in future research.

Totally, government in various countries or regions should formulate more targeted policies to reduce the incidence of facial fractures. For example, attention should be

paid more on traffic control in regions with high SDI whilst in regions with low SDI, improvement of diagnosis ability of facial fractures should be more concerned. It is also essential to promote relevant protection awareness on entertainments with high risk in public, especially among young males. Moreover, the enormous health burden of facial fractures caused by warfare should not be neglected. With the increasing pattern of life expectancy globally, elderly people should be more focused on.

## ACKNOWLEDGEMENTS

We appreciated the works by the Global Burden of Disease 2017 study collaborators as well as all individuals who contributed to find, catalogue and analyze the detailed data about human diseases, which dramatically facilitated the public health communication worldwide.

### Funding

This work was supported by the National Natural Science Foundation of China (NO.81702708), the Natural Science Foundation of Hunan Province (NO. 2017JJ2392 and NO. 2018JJ3862), the Scientific Research Project of Hunan Provincial Health Commission (NO. B20180054), and the Changsha Science and Technology Project (NO. kq1706072). The funders had no role in study design, data collection and analysis, decision to publish, or preparation of the manuscript.

### Grant Disclosures

The following grant information was disclosed by the authors:
National natural science foundation of China: 81702708.
Natural Science Foundation of Hunan Province: 2017JJ2392, 2018JJ3862.
Scientific Research Project of Hunan Provincial Health Commission: B20180054.
Changsha Science and Technology Project: kq1706072.

### Competing Interests

The authors declare there are no competing interests.

### Author Contributions

- Jin Wu analyzed the data, prepared figures and/or tables, and approved the final draft.
- Anjie Min conceived and designed the experiments, prepared figures and/or tables, and approved the final draft.
- Weiming Wang performed the experiments, authored or reviewed drafts of the paper, and approved the final draft.
- Tong Su conceived and designed the experiments, authored or reviewed drafts of the paper, and approved the final draft.

### Data Availability

   Raw data are available in the Supplemental Files.

## Supplemental Information

Supplemental information for this article can be found online at http://dx.doi.org/10.7717/peerj.10693#supplemental-information.

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
