# Peer review of "Trends in the incidence, prevalence and years lived with disability of facial fracture at global, regional and national levels from 1990 to 2017"

_PeerJ, doi:10.7717/peerj.10693_

## Round 0.1 · original submission · Major Revisions

Dear Authors,

Your manuscript has been reviewed by three experts and all have provided comments to improve the manuscript. Please find their comments.

Sincerely,

Gunjan

Reviewer 1 ·

Basic reporting

Authors have used clear and professional English throughout the manuscript. Authors have provided proper literature references. However, I think they could have elaborated more on the background/context. They could have emphasized why the study is important in more details. Article structure is professional. Figure legends could be made bit more legible. Tables are very professional. The manuscript is self-contained with relevant results to hypothesis.

Experimental design

The experimental design is standard as expected in the field. The research question is well defined. Author have collected data from legitimate sources and performed proper statistical analyses. Methods have been described with sufficient details.

Validity of the findings

The findings of the manuscript are valid and will definitely help policymakers around the world. I really liked the way authors discussed the results and put it in the context.

Reviewer 2 ·

Basic reporting

The manuscript is written in professional, Scientific English. The introduction just sufficiently sets the theme of the study and is well-referenced.
However, all the figures and tables miss the description and headings which makes it very difficult to follow. Please label them.
Fig-1 The legend is not clear
Fig-2 What is the difference between graphs 2A, 2B and 2C?
Fig-3 What is the difference between graphs 3A, 3B and 3C?
Fig-4 Labels A,B, C missing.
Fig-5 Please explain how the correlation between EAPC and SDI is different among Fig. 5B, 5D and 5F.
What is the difference between Fig-1 and Fig-6? They look exactly the same.
Fig-7 misses description.
Fig-8 Which trend does each graph represent?
Fig-9 Fig 9B and 9C are the same.
Fig-10 misses description.

Experimental design

The research question is well-framed. Though the data source is mentioned in the abstract, the methods section is missing. Thank you for providing the raw and processed data. Could you please:
1. Describe the methodology used to analyze the data and draw statistical inference for YLDs, SDI, and APRs?
2. How was EAPC calculated?
3. Why did you choose data only till 2017? Although the latest 2019 report of Global Burden of Disease is available.

Validity of the findings

The study is exhaustive and provides all underlying data. The study is presented in a tabulated form region-wise, making it easier to follow.
Please state explicitly how this study would directly impact the medical resource allocation and policymakers as stated in the introduction (provide if you have any correlation data).

Additional comments

No comments

Reviewer 3 ·

Basic reporting

The figures should be made in a more representable form.

Experimental design

The methodology section is missing throughout the manuscript.

Validity of the findings

No comments

Additional comments

The manuscript “Trends in the incidence, prevalence, and YLDs of facial fracture at global, regional, and national levels from 1990 to 2017” is a good compilation of data from 1990 to 2017 to understand the overall health burden of facial fractures. The study could be useful to the countries to make policies to reduce the occurrence of these events in the society. The study provides a thorough in-depth analysis of the available data. However, there are few concerns which needs to be addressed:

General Comments:
1. The author should make a separate section for describing the method used. This section should be separate from method described in the abstract.
The authors should not use an acronym in the manuscript title. The authors have extensively used acronyms throughout the manuscript, most of the time it makes it difficult for the reader to follow. Overall, the authors need to use less acronyms.
2. The text of the manuscript has some formatting errors.
Line 254 “this first study was the to systematically¨
Line 229 “leading to a high death rate in (Toivari et al. 2016)
Line 62 and 71 GBD acronym should be introduced in line 62 rather than 71

3. In addition, the authors need to make figures in a more representable form for better understanding. The text of the labels needs to be enlarged For eg.; figure 2 need to be magnified.

Specific comments:

1. The full form of ASIR has not been mentioned in the manuscript but used frequently by the authors in the complete manuscript.
2. The labels of figure 4 are absent.
3. The legends of figure 9 does not match with the figures. Authors should check these errors.

---

## Round 0.2 · accepted · Accept

Dear Dr. Wu and Dr. Wang,

Your manuscript is evaluated by two independent reviewers and based on reviewers' response to the revised manuscript I am pleased to inform you that your manuscript is accepted.

Sincerely,

Gunjan

Reviewer 3 ·

Basic reporting

no comment

Experimental design

no comment

Validity of the findings

no comment

Additional comments

no comments

Reviewer 4 ·

Basic reporting

The manuscript is well written. The results and conclusions of the papers are interesting and convincing. I also find the analyses to be relevant and well documented. Statistical analysis is thorough and reliable.

Experimental design

The manuscript is well designed.

Validity of the findings

No comments.

Additional comments

Dear Authors,
Congratulation for this interesting study. The manuscript by Wu et al. reported the systematic assessment of global burden of facial fractures across all age and sex group. I have some suggestions for the authors.

1. The authors may consider to add a table depicting sources of clinical records used calculating cause-nature proportions for facial fractures.
2. What is the basis of selecting the 1990 as starting year?
3. This study reported that- the ratio of facial fractures between males and females was 2:1. What could be the reason?